# Validation of FUNMOVES: A reliable tool for assessing motor skills in Spanish schoolchildren

**Pablo Lizoain**[1], **Diana Rodriguez Romero**[1], **Celeste Reyes-Vivanco**[1], **Nick Preston**[2,3], **Lucy H. Eddy**[2,4,5], **Liam J. B. Hill**[2,6], **Sara Magallón**[1]☉*, **Martín Martínez**[1]☉*

1 Education and Psychology faculty, University of Navarra, Pamplona, Spain, 2 Centre for Applied Education Research, Wolfson Centre for Applied Health Research, Bradford Royal Infirmary, Bradford, West Yorkshire, United Kingdom, 3 Academic Department of Rehabilitation Medicine, University of Leeds, Leeds, United Kingdom, 4 Department of Psychology, University of Bradford, Bradford, United Kingdom, 5 Bradford Institute for Health Research, Bradford Royal Infirmary, Bradford, United Kingdom, 6 Moray House School of Education and Sport, University of Edinburgh, Edinburgh, United Kingdom

☉ Equal contribution.
* mmvillar@unav.es (MM); smagallon@unav.es (SM)

## Abstract

### Background

Developmental Coordination Disorder (DCD) affects around 5% of the global population, but difficulties in accessing specialist diagnoses lead to underdiagnosis. In Spain, there are no school-based screening protocols for motor coordination difficulties with published evidence of reliability and validity, and specialist assessment is only available through overburdened healthcare services. Despite its prevalence, DCD often goes undetected due to limited access to assessments and the lack of easy-to-use tools in educational settings.

FUNMOVES was developed to assess fundamental motor skills (FMS) and has shown promising signs of validity and reliability in the UK. It allows entire classes to be screened during a single Physical Education (PE) session, helping identify children with motor difficulties.

### Aim

To adapt FUNMOVES to the Spanish school context and evaluate its psychometric properties and diagnostic accuracy when used by PE teachers.

### Methods

FUNMOVES was translated and culturally adapted for Spain. Structural validity was assessed using Rasch analysis with data from 243 children. Convergent validity was evaluated in a subsample of 50 children by comparing FUNMOVES with the MABC-2, the gold-standard motor assessment.

**Data availability statement:** All relevant data are within the paper and its Supporting Information files.

**Funding:** Several authors received external funding supporting their research contracts: P. Lizoain is supported by the grant PRE2021-097858, funded by the Ministerio de Ciencia e Innovación (MCIN) / Ministry of Science and Innovation (Spain) and the Agencia Estatal de Investigación (AEI) / State Research Agency (Spain), and co-funded by the European Social Fund Plus (ESF+ U): MCIN/AEI/ 10.13039/501100011033. L.H. Eddy is supported by a grant from the Waterloo Foundation (ref: 27665413). S. Magallón is supported by the Ramón y Cajal grant RYC-2017-22060, funded by the Ministerio de Ciencia e Innovación (MCIN) / Ministry of Science and Innovation (Spain) and the Agencia Estatal de Investigación (AEI) / State Research Agency (Spain), and by the project PID2020–119328GA-I00 (AEI Proyectos I+D+i, funded by MCIN/AEI). The funding sources had no role in study design, data collection and analysis, decision to publish, or preparation of the manuscript.

**Competing interests:** The authors have declared that no competing interests exist.

## Results

Rasch analysis confirmed the unidimensionality and local independence of the Spanish FUNMOVES, with good model fit. Some item thresholds were initially disordered but were corrected through rescoring. Using the MABC-2 10th percentile in total score as the criterion for motor impairment, FUNMOVES showed moderate accuracy (AUC = 0.78), with 80% sensitivity and 68% specificity at the 17th percentile cut-off. It also showed a high negative predictive value (0.89). Broadening the diagnostic criteria did not significantly improve accuracy.

## Implications

The Spanish FUNMOVES shows initial structural and convergent validity for school-based FMS screening. Its simplicity, low cost, and scalability support early detection and intervention. The tool is freely available and can be implemented using existing school resources in one PE session, potentially accelerating the diagnostic process.

## 1 Introduction

The Fundamental Motor Skills (FMS) are foundational to children's motor development. They are typically classified into three core domains: locomotor development (e.g., walking, jumping, moving), object control manipulation (e.g., writing, throwing and catching objects, or tying shoelaces), and balance (static or dynamic) [1–3]. Adequate development of these skills is essential not only for daily functioning but also for broader outcomes, including academic performance and socio-emotional well-being [4–8]

Developmental Coordination Disorder (DCD) is a neurodevelopmental condition characterized by difficulties in acquiring and executing coordinated motor skills (a). These impair daily activities and academic performance (b), symptoms appear in the early stages of development (c) and (d) it cannot be explained by visual or intellectual disabilities [9,10]. Its global prevalence is estimated between 2% and 8% [9] with studies reporting a prevalence of approximately 9% in the Spanish school-age population [11]. DCD frequently co-occurs with other neurodevelopmental disorders, such as Attention Deficit Hyperactivity Disorder, Autism Spectrum Disorder, and specific learning disorders, highlighting the importance of comprehensive assessment [12–15]. Motor difficulties often persist into adolescence and adulthood. They affect academic and occupational functioning, and also impair social relationships due to limited peer participation and reduced self-esteem [16].

Additionally, there is evidence that regular physical activity performed at moderate intensity provides physical and mental health benefits to individuals [17–21]. However, individuals with DCD engage in less physical activity compared to their peers [22,23]. This can lead to problems and challenges in both physical well-being (e.g., obesity, diabetes, or cardiovascular diseases) [24–27] and mental health, such as anxiety, depression, or difficulties in social relationships [28–30].

Therefore, there is a clear need for early diagnosis to enable interventions for children with DCD [31–33]. Currently, difficulties in FMS are only assessed if families consult a pediatrician, who then refers them to a specialist [34,35]. In Spain, long waiting times and limited specialist access delay diagnosis and intervention [36,37]. Moreover, motor difficulties are rarely perceived by families as a primary concern compared to challenges in literacy or numeracy, and are often underestimated or overlooked [38,39]. As a result, motor clumsiness is seldom the main reason for seeking medical advice, leading to underdiagnosis of this disorder [40–42]. This underscores the importance of validating screening tools like FUNMOVES within the Spanish context, to facilitate early identification in school settings and bridge the gap in access to specialized care.

Given the current context, it is essential to identify tools that allow for the universal screening of motor skill difficulties in children. Schools serve as a key setting for efficiently and systematically screening children's FMS, as they spend a significant portion of their daily schedules there. Additionally, the Physical Education class presents an ideal context for evaluating motor skills. Thus, school-based evaluation offers a necessary solution for identifying children with motor difficulties and provides a promising collaborative framework that could help reduce healthcare waiting lists and the need for family consultations.

The MABC-2 is one of the most widely used tools for evaluating the first criterion of the DSM-5 diagnostic framework [31]. This refers to motor skill acquisition and execution being significantly below expectations for the individual's age and learning opportunities. This battery includes different activities for three age groups ranging from 4 to 16 years old. It is administered individually and takes approximately 30–40 minutes to complete [43]. Although a third edition of the MABC is available, it has not yet been validated in the Spanish population. This justifies the continued use of the second edition (MABC-2) in research and clinical practice within this context [44,45]. There is also the Bruininks-Oseretsky Test of Motor Proficiency (BOT), 3rd Edition, which allows for the assessment of individuals aged 4–26 years in approximately 50–90 minutes. This test can separately examine fine motor skills, gross motor skills, and digital dexterity [46]. With regard to screening tools for the detection of motor difficulties, several instruments are currently available. These include the Developmental Coordination Disorder Questionnaire (DCDQ), designed for completion by parents; the Motor Observation Questionnaire for Teachers, which, however, is not yet available for use with Spanish populations; and the Movement Assessment Battery for Children (MABC) teacher checklist, intended for use in classroom settings. [47–49].

In terms of diagnostic accuracy, these screening tools show varying levels of performance. This performance aligns with the upper band typically reported for school-based screening questionnaires pooled AUC ≈ 0.80 for the parent-completed DCDQ according to a recent meta-analysis [50] and is comparable to teacher tools such as the Motor Observation Questionnaire for Teachers, which reaches an AUC of roughly 0.73 [51]. By contrast, very brief observational checklists like the original MABC teacher checklist offer high specificity but low sensitivity, yielding an effective AUC close to 0.55 [52].

For the feasible assessment of FMS in school environments, Klingberg et al. propose that evaluation tools meet specific criteria to ensure practicality and adaptability [53]. The assessment should be efficient, taking no longer than 10 minutes per child, and should utilize a maximum of six tasks to maintain simplicity. Additionally, the materials used must be part of the common inventory typically available in schools, avoiding the need for specialized resources, while the required space should not exceed 6 m² per person, ensuring compatibility with school settings. The recording of the assessment should focus on the final product rather than the process to facilitate straightforward evaluation. Moreover, school staff should be able to carry out the assessment autonomously, given that the training required for administration lasts less than one hour, facilitating its accessibility and integration into daily routines.

A systematic review led by Eddy et al. [2020] demonstrated that most of the tools available at the time, such as the MABC-2 and BOT batteries, did not meet the criteria for school-based assessment. These tools were considerably expensive and required a per-student time commitment well beyond the proposed 10-minute limit. Other tools that were more feasible for school environments lacked valid evidence. Considering this context, Eddy and his team proposed the

development and validation of FUNMOVES that would meet the outlined criteria and be suitable for implementation in schools [54,55]. The original tool development reported structural validity evidence: a Person Separation Index (PSI) of 0.64, unidimensionality (5.36% significant tests; 95% CI = .02,.09), and good fit model ($\chi^2(14) = 20.42$, p = 0.12) [55]. When evaluating FUNMOVES total score against the MABC-2 total score, the specificity (1, 95%CI = 0.63−1) and positive predictive value (1; 95%CI = 0.68−1) of FUNMOVES were high, whereas sensitivity (0.57, 95%CI = 0.29–0.82) and negative predictive values (0.57, 95%CI = 0.42–0.71) were moderate. Since FUNMOVES does not account for manual dexterity, a new analysis was conducted excluding this subscale. When using a broader criteria for evaluating only MABC-2 subscales which are directly related to FMS (Aiming & Catching, and Balance) these values improved to 0.89 (95% CI = 0.52−1) and 0.93 (95% CI = 0.67–0.99) respectively [56].

Current evidence suggests FUNMOVES is a psychometrically robust tool for assessing FMS in UK primary school students. The construct assessed by FUNMOVES is unidimensional. Its scores show adequate internal consistency and can distinguish between skill levels and age groups. Therefore, its use is feasible in schools, as it is freely available. Moreover, it meets the criteria established by Klingberg et al. [53]. A full class can be assessed in a single PE session (criteria i); the test involves six activities using commonly available PE equipment (criteria ii and iii). These activities take place on a 5x5 m² grid painted on the floor (criteria iv). They are led by the PE teacher after less than one hour of training, and performance is recorded based on the final product of each participant's execution (criteria v, vi, and vii).

The FUNMOVES test involves performing a series of activities in which all students in a class participate in groups of five. These tasks encompass various motor skills, including running, jumping, throwing, kicking, and balance (Table 1).

This study aims to adapt FUNMOVES for use in the Spanish education system. It also examines its psychometric properties and effectiveness when administered by PE teachers. Furthermore, the tool is intended as part of a structured diagnostic pathway. Screening could begin in PE classes, followed by data evaluation by school guidance teams, and referral for clinical diagnosis when needed.

All participants will be assessed using the Spanish version of FUNMOVES. Subsequently, a subsample of participants will be evaluated using the MABC-2 battery. For the validation conducted with a Spanish population, similar results to those obtained in the English population are expected [55,56]. Finally, the study involving the Spanish sample will be

**Table 1. Description and activities score of FUNMOVES.**

| Item | Activity | Score |
|---|---|---|
| Running | Run from the first line to the last one as many times as possible on 15". | (0–10) Number of lengths completed. |
| Jumping | Making small jumps go from the first line to the last one, stopping 3" on the lines of the grid. | (1–4) The zone where the balance is lost by participants. |
| Hopping | Making small hops go from the first line to the last one, stopping 3" on the lines of the grid. | (1–4) The zone where the balance is lost by participants. |
| Throwing | Throwing, underarm, 5 beanbags, trying to leave 1 in each box. This activity is completed twice (right and left arm). | (0–5) Number of boxes filled with a beanbag. |
| Kicking | Kicking, 5 beanbags (1 each time), trying to leave 1 in each box. | (0–5) Number of boxes filled with a beanbag. |
| Balance | Participants must hold these balances:<br>1. Stand, while passing a beanbag around their body 3 times.<br>2. Stand on one leg, while passing a beanbag around their body 3 times.<br>3. Stand on one leg while picking up a beanbag from the floor.<br>4. Stand on one leg, eyes closed, while passing a beanbag around their body 3 times. | (0–4) Balance achieved without falling. |

*Note: Activities of the FUNMOVES Manual in the UK.*

complemented by a Receiver Operating Characteristic (ROC) analysis, where an Area Under the Curve (AUC) of ≥ 0.70 is predicted [57].

## 2 Methods

### 2.1 Study design

This study employs a cross-sectional design with intentional sampling. First, FUNMOVES was translated and adapted to the Spanish context. Thereafter, all participants underwent assessment using the Spanish adaptation of the FUNMOVES tool. Following this, a subsample of individuals who demonstrated low performance on FUNMOVES—classified as probable Developmental Coordination Disorder (pDCD group)—and another subsample with typical performance—classified as typically developing (TD group)—were further evaluated using the MABC-2 assessment battery. The purpose of the MABC-2 assessment was to objectively address the first criteria of the DSM-5 and assess the predictive validity of FUNMOVES.

### 2.2 Recruitment

For recruitment, PE teachers, school administrators, and principals from various schools in Navarra (Spain) were contacted. The process began with phone and email contact to present the project. This was followed by meetings with school staff to explain objectives and schedule evaluations, and sessions with families to distribute consent forms.

Data were collected from children whose parents or legal guardians provided written informed consent authorizing participation in the FUNMOVES assessment, conducted between 30/01/2023 and 31/03/2023. An additional written informed consent form was required from the same parents or guardians for those participating in the MABC-2 evaluation, which took place between 25/10/2023 and 23/11/2023. All consent forms were signed and securely stored. The study was approved by the Ethics Committee of the University of Navarra (reference: 2022.088). No waiver of consent was granted.

### 2.3 Participants

Parents of 316 children provided informed consent for their participation in the group-based FUNMOVES assessment. Of these 316 children, data from 73 were excluded from the final sample due to various reasons: absence on the day of the assessment, temporary injury preventing participation, or procedural issues during administration by the teacher (children practicing, poor demonstrations, and lack of clarity in instructions). As a result, a total of 243 participants (mean age = 9.5 years, sd = 1.38) were assessed using the FUNMOVES tool. The final sample included 134 boys (55.14%) and 109 girls. Teachers reported each student's dominant hand (based on writing) and noted suspected motor difficulties (right-handed: 92.59%; suspected difficulties: 6.59%) (Table 2).

A subsample of sixty participants were invited for the individual assessment using the MABC-2. They were assigned to two groups based on their FUNMOVES development: thirty children from the pDCD group and thirty children from the TD group, matched by age with the evaluation group. The pDCD and TD groups were not matched by sex, as the MABC-2 manual does not provide sex-based norms. Of the sixty invited participants, the parents of fifty children provided informed consent: twenty-four in the pDCD group and twenty-six in the TD group (Fig 1).

### 2.4 Procedure

The initial phase of the process involved translating the FUNMOVES manual into Spanish. The translated version was subsequently reviewed by the research team in collaboration with PE teachers, who provided feedback leading to necessary adaptations. Modifications included translation, reformatting of the manual, and the addition of practical classroom management tips to support effective implementation, followed by a training presentation with explanatory task videos to support teacher training. The final version of the adapted FUNMOVES manual is available as Supporting Information

**Table 2. Demographic description of participants in the FUNMOVES evaluation.**

| Year | Sex Boys (%) | Girls | Age (years) Mean | sd | Dominant hand Right (%) | Left | Suspected difficulties No (%) | Yes |
|---|---|---|---|---|---|---|---|---|
| 1 | 8 (66.67) | 4 | 6.65 | 0.30 | 10 (83.33) | 2 | 12 (100) | 0 |
| 2 | 13 (59.09) | 9 | 7.67 | 0.35 | 21 (95.45) | 1 | 16 (72.73) | 6 |
| 3 | 48 (53.93) | 41 | 8.69 | 0.29 | 80 (89.89) | 9 | 86 (96.63) | 3 |
| 4 | 5 (33.33) | 10 | 9.72 | 0.42 | 14 (93.33) | 1 | 12 (80) | 3 |
| 5 | 47 (54.02) | 40 | 10.70 | 0.31 | 82 (94.25) | 5 | 82 (98.80) | 1 |
| 6 | 13 (72.22) | 5 | 11.65 | 0.35 | 18 (100) | 0 | 15 (83.33) | 3 |
| **Total** | **134 (55.14)** | **109** | **9.50** | **1.38** | **225 (92.59)** | **18** | **223 (91.77)** | **16 (6.59)** |

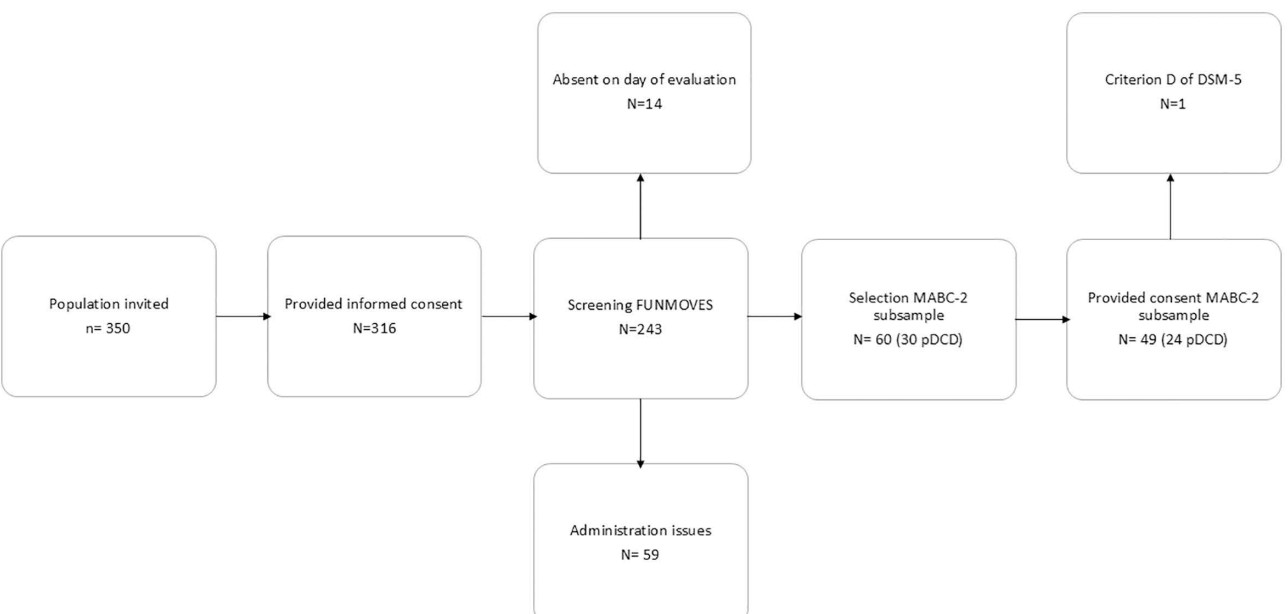

**Fig 1. Participant flow diagram.** Flow diagram showing the recruitment process, exclusions, and formation of the FUNMOVES and MABC-2 subsamples.

under a Creative Commons Attribution (CC BY) license, allowing unrestricted use, distribution, and reproduction in any medium, provided the original work is properly cited.

Once the final version of the manual was completed, PE teachers from the participating schools received training on how to implement the tool and how to record student data for the study from the research team. Two researchers supported the teaching staff during test implementation. PE teachers recorded performance data in a spreadsheet and submitted it to the research team after completing all evaluations. Additionally, printed score sheets used during the assessments were also provided by the PE teachers.

After analyzing the FUNMOVES performance data, the researchers conducted the MABC-2 assessment in a classroom during school hours, with permission from school principals and teachers. These assessments were conducted blindly, as the evaluators were not informed of the participants' FUNMOVES performance during the initial phase.

## 2.5 Data analysis

The same validation procedure as described in the original FUNMOVES study was followed, including Rasch analysis and concurrent and predictive validity analysis [55,56].

### 2.5.1 Structural validity.

The Rasch model, a probabilistic method from Item Response Theory, was used to assess the tool's psychometric properties. It evaluates whether a total score can be derived from the sum of individual item scores (i.e., FUNMOVES tasks). In this model, the probability of a correct response to an item is a function of the individual's ability and the item's difficulty. Rasch analysis allows for the evaluation of item fit to a common construct, detection of local dependence (item redundancy), and assessment of how item score structures impact measurement validity. When responses fit the model, both items and individuals can be placed on a logit scale, facilitating their comparison. On this scale, items with higher logit values are more difficult, and individuals with higher logit values represent higher ability levels [58–60].

The analyses were conducted using the Partial Credit Model without constraints, implemented in RUMM2030 software, due to differences in response category structures. The model fit was assessed through a statistical summary, including the mean location of individuals and items, and a chi-square test. A non-significant chi-square indicates good fit, meaning items function consistently across ability levels.

The PSI, was used to assess internal consistency and the ability of the scale to distinguish between performance levels. A PSI ≥ 0.70 is acceptable and a PSI ≥ 0.50 can discriminate between two levels [58,59]. Individual item fit was examined to verify conformance to the model, ordered thresholds, and the absence of differential item functioning or local dependence.

Principal Components Analysis (PCA) of the residuals was used to test for unidimensionality, with paired t-tests conducted between subsets of items. Unidimensionality was assumed when fewer than 5% of these t-tests were statistically significant [61]. Items showing poor fit or local dependence were candidates for removal. Disordered thresholds were addressed by collapsing response categories where appropriate.

### 2.5.2 Concurrent and predictive validity.

To select participants for the MABC-2 assessment, FUNMOVES total scores were used to compute percentiles relative to classmates in the same academic grade. Percentile scores were also calculated based on the norms provided in the MABC-2 manual, normalized for the Spanish population. These were obtained for each of the three MABC-2 subscales and the total score. Based on the MABC-2 scores, two operational criteria were established to define potential cases of motor impairment. The Strict criterion considered only the MABC-2 total score below the 10th percentile, representing a more specific and restrictive definition that focuses exclusively on overall motor coordination. In contrast, the Broad criterion classified a child as positive if either the total score or any of the subscale scores (Aiming & Catching or Balance) fell below the 10th percentile. This broader approach captures partial motor coordination difficulties and therefore includes a wider range of potential positive cases.

Concurrent validity (agreement between FUNMOVES and MABC-2 scores) and predictive validity (classification accuracy of FUNMOVES) were assessed using standard diagnostic metrics: sensitivity, specificity, positive predictive value (PPV), negative predictive value (NPV), and overall accuracy.

Additionally, a ROC analysis was performed to estimate the diagnostic accuracy of FUNMOVES for identifying motor impairment as defined by the MABC-2. ROC curves and the AUC were computed with the pROC package [62]. Ninety-five-percent confidence intervals (CIs) for AUC were obtained via a 2,000-sample stratified bootstrap [63]. The optimal threshold was chosen with Youden's J (sensitivity + specificity − 1) [64]. Paired AUCs were compared by DeLong's test [65].

Statistical analyses for evaluating concurrent and predictive validity were performed using R and RStudio. A p-value of 0.05 was set as a significance threshold.

# 3 Results

## 3.1 Structural validity

The initial Rasch analysis indicated that FUNMOVES is a unidimensional tool with only 0.82% of the paired t-tests showing significant differences. The data showed good overall model fit ($\chi^2$(21) = 21.27, p = 0.44), and acceptable internal consistency (PSI = 0.53).

All items demonstrated good individual fit to the model, were locally independent, and showed no evidence of differential item functioning or bias. However, some items -such as running, jumping and hopping, throwing with the dominant hand, and balance- presented disordered thresholds in one or more response categories. Accordingly, a rescoring of these activities was proposed to restore threshold order (see Table 3).

Once the activity categories were rescored, the new Rasch analysis (Table 4) confirmed that FUNMOVES remains a unidimensional instrument, with only 4.53% of the paired t-tests showing significant differences. The data continued to demonstrate a good fit to the Rasch model ($\chi^2$(21) = 14.6, p = 0.84), with acceptable internal consistency (PSI = 0.52).

The rescoring of item categories restored the ordered structure of thresholds (Fig 2, right). Additionally, the person-item distribution map supported the interpretation that this PSI value is suitable for identifying individuals with low motor proficiency (Fig 3). All items once again showed good individual fit, local independence, and no evidence of bias.

## 3.2 Concurrent and predictive validity

Fifty participants were assessed using the MABC-2 scale, of whom 26 were part of the TD group and 24 belonged to the pDCD group. In both the subscales and the total score of the MABC-2, participants in the pDCD group performed significantly worse than those in the control group (Table 5). In addition, a significant difference was observed between groups

**Table 3. Rescoring of FUNMOVES items based on Rasch analysis results.**

| Item | Rasch analysis issue | Modification applied |
|---|---|---|
| Running | Disordered thresholds: categories 1–5 were never more likely than 0. | Combine the first five categories into a single category. |
| Jumping | Disordered thresholds: categories 2–3 were never more likely than others. | Combine the first three categories into a single one (dichotomous: all-or-nothing). |
| Hopping | Disordered thresholds: categories 2–3 were more likely than categories 1 or 4. | Combine categories 2 and 3. |
| Throwing DOM | Disordered thresholds: categories 1–2 were never more likely than 0. | Combine the first three categories. |
| Throwing NDOM | No issues identified. | No modification needed. |
| Kicking | No issues identified. | No modification needed. |
| Balance | Disordered thresholds: category 3 was never more likely than category 4. | Combined the last two categories. |

**Table 4. Descriptive statistics of the Rasch analysis.**

| Description | Item location | | Person location | | Item fit residual | | Person fit residual | | Chi-square interaction | | | Person separation index | Unidimensionality | | |
|---|---|---|---|---|---|---|---|---|---|---|---|---|---|---|---|
| | Mean | SD | Mean | SD | Mean | SD | Mean | SD | Value | df | p | | Num. Sig. test | Out of | % |
| Initial analysis | 0 | 1.38 | 1.09 | 0.63 | 0.15 | 0.8 | −0.18 | 0.78 | 21.27 | 21 | 0.44 | 0.53 | 2 | 243 | 0.82 |
| Rescored | 0 | 1.37 | 1.14 | 0.77 | 0.18 | 0.38 | −0.16 | 0.9 | 14.6 | 21 | 0.84 | 0.52 | 5 | 243 | 4.53 |

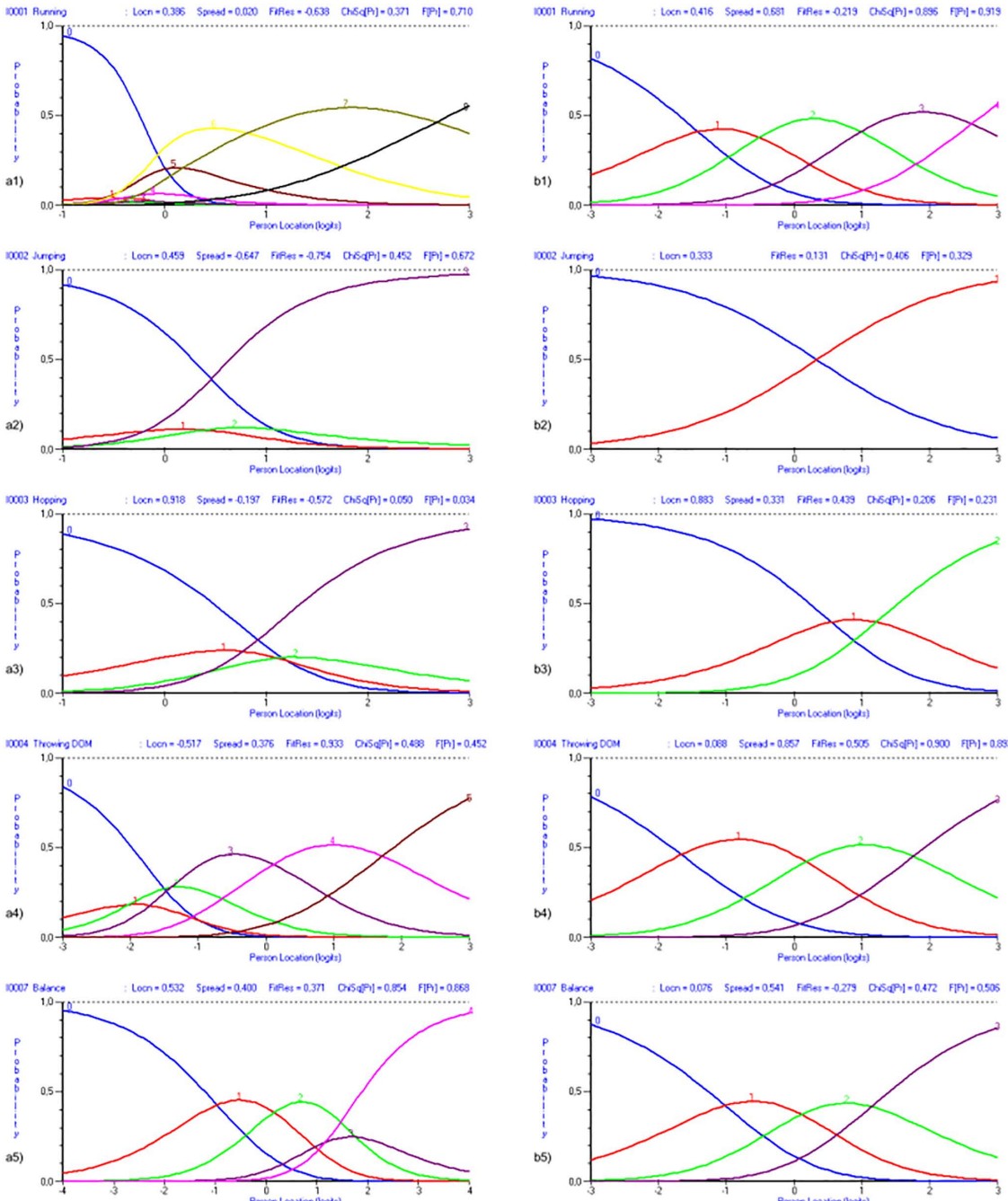

**Fig 2. Category probability curves (a1) running; b1) running rescored; a2) jumping; b2) jumping rescored; a3) hopping; b3) hopping rescored; a4) throwing DOM; b4) throwing DOM rescored; a5) balance; b5) balance rescored).** The left column (a) contains the curves for each activity in the initial analysis. The right column (b) shows the category curve for the same item, but in the rescored analysis.

in the variable teacher suspicion (p = 0.011), with a higher proportion of suspected cases in the pDCD group. This result was expected, as children classified as pDCD based on FUNMOVES (scores below the 10th percentile or FUNMOVES scores below the 15th percentile plus teacher suspicion) are more likely to have been previously identified because of

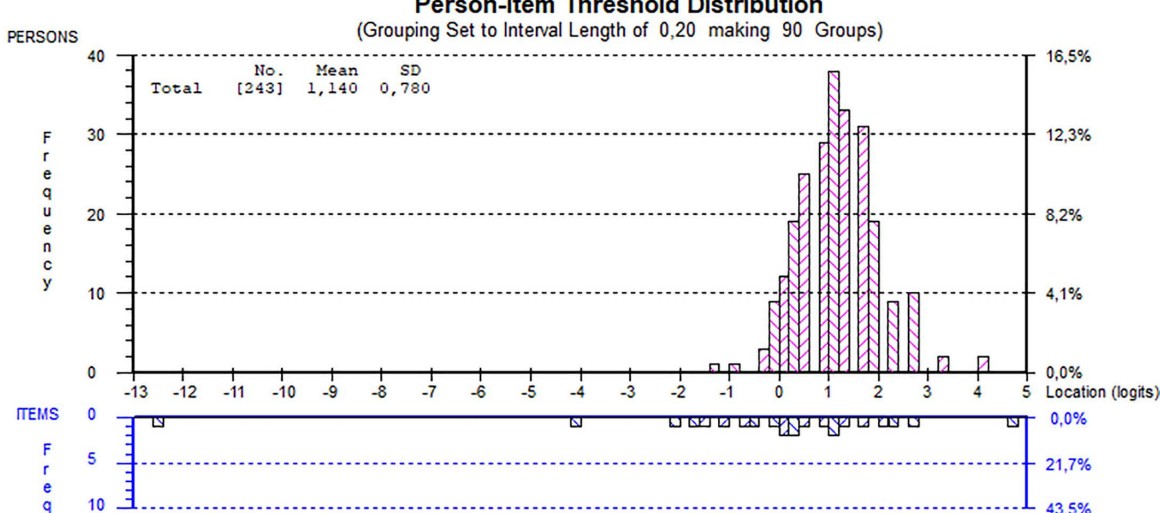

**Fig 3. Table distribution of item difficulty and person abilities for rescored analysis.**

**Table 5. Description of the sample assessed using MABC-2.**

| | | TD | pDCD | p-value* |
|---|---|---|---|---|
| **Sex** | Boys | 11 | 15 | 0.312 |
| | Girls | 14 | 9 | |
| | Total | 25 | 24 | |
| **Dominant Hand** | Right | 25 | 24 | NA |
| | Left | 0 | 0 | |
| **Age** | Mean (SD) | 9.14 (1.55) | 9.23 (1.63) | 0.843 |
| **MABC-2 (percentile)** | MABC-2 Total | 45.00 | 15.42 | <0.001 |
| | MABC-2 Manual dexterity | 41.68 | 20.58 | 0.003 |
| | MABC-2 Aiming and catching | 38.08 | 32.71 | 0.319 |
| | MABC-2 Balance | 60.12 | 28.25 | <0.001 |
| **Suspected difficulties** | | 1 | 9 | 0.011 |

*Note: * Chi squared test used for: sex and suspected difficulties. T-test used for: age (normal distribution). Wilcox-test used for MABC Total and sub-scales.*

the definition of the group pDCD. One participant from the control group had to be excluded based on criterion D of the DSM-5.

Fig 4a and 4b presents the confusion matrix providing the number of true positives, false positives, false negatives, and true negatives. For the calculation of this matrix, participants assessed with FUNMOVES are considered positive when scoring below the 10th percentile or the 15th percentile, plus the teacher's suspicion of motor difficulties. In the case of the MABC-2, two criteria were used to classify participants as positive: (i) scoring below the 10th percentile in the total score of the test (i.e., strict criteria), and (ii) scoring below the 10th percentile in the total score or scoring below the 10th percentile in Aiming & Catching or Balance MABC-2 subscales (i.e., broad criteria).

To assess FUNMOVES as a screening tool, sensitivity, specificity, PPV, NPV, and accuracy were calculated for overall motor skills and each domain (Table 6).

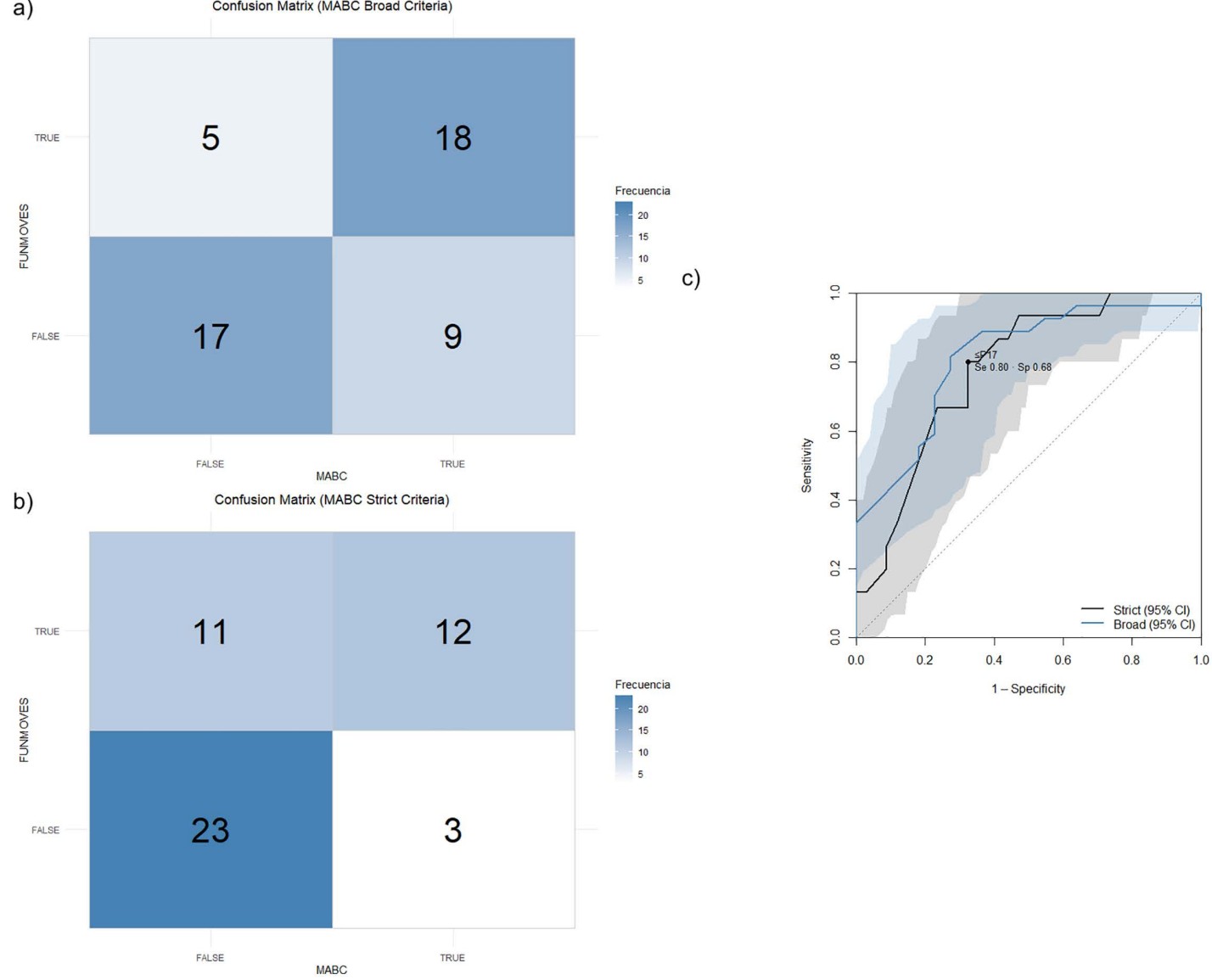

**Fig 4. a) Confusion matrix using broad criteria.** b) Confusion matrix using strict criteria. c) ROC curves for FUNMOVES percentiles in detecting motor impairment defined by the MABC-2. 17th percentile cut-off for strict criteria. The black curve corresponds to the strict definition of impairment (MABC-2 total < 10th percentile) and the blue curve to the broad definition (total or Aiming & Catching or Balance < 10th percentile). Shaded areas represent 95% bootstrap confidence bands (2,000 replicates).

As shown in Fig 4c, the ROC analysis showed that FUNMOVES discriminated moderately well between children with and without motor impairment. Under the strict MABC-2 criterion (total score), the AUC was 0.78 (95% CI 0.64–0.90). The optimal cut-off was the 17th FUNMOVES percentile, which balanced the trade-off at 80% sensitivity and 68% specificity (Youden J = 0.48). When the broader criterion was applied (total score or Aiming&Catching or Balance subscales), the AUC rose slightly to 0.81 (95% CI 0.69–0.92). The best cut-off in this case was the 35.5th percentile, achieving 82% sensitivity and 73% specificity (Youden J = 0.54). A paired DeLong test confirmed that the difference between the two AUCs was not statistically significant (p = 0.71), indicating that the broader diagnostic definition did not meaningfully enhance the overall accuracy of FUNMOVES.

**Table 6. Concurrent and predictive validity table of FUNMOVES and MABC-2.**

| | Strict criteria* | | Broad criteria** | |
|---|---|---|---|---|
| | **Estimate** | **Range CI** | **Estimate** | **Range CI** |
| **Sensitivity** | 0.80 | 0.55-0.93 | 0.67 | 0.48-0.81 |
| **Specificity** | 0.68 | 0.51-0.81 | 0.77 | 0.57-0.90 |
| **PPV** | 0.52 | 0.33-0.71 | 0.78 | 0.58-0.90 |
| **NPV** | 0.89 | 0.71-0.96 | 0.65 | 0.46-0.81 |
| **Accuracy** | 0.71 | 0.58-0.82 | 0.71 | 0.58-0.82 |

Note: *MABC positive < 10th on MABC Total. **. MABC positive < 10th on MABC-2 Total OR < 10th on Aiming&Catching or Balance MABC-2 subscale.

## 4 Discussion

From a psychometric perspective, Rasch analysis confirmed that FUNMOVES exhibits a unidimensional structure, supporting its theoretical coherence as a global measure of motor competence. This unidimensionality implies that the tool assesses a single latent trait, which is desirable for screening purposes. However, the person separation index (PSI ≈ 0.52) was relatively low, indicating limited ability to distinguish between different levels of motor proficiency, particularly in the mid-range of the ability spectrum. While this may constrain its use for fine-grained diagnostic classification, it remains acceptable for identifying children with significant motor difficulties, the primary goal of a screening tool. Several items initially displayed disordered thresholds, prompting a rescoring of response categories to restore ordinal structure. This adjustment improved the interpretability of item responses and preserved model fit. Furthermore, the absence of differential item functioning (DIF) and the confirmation of local independence among items reinforce the robustness and fairness of the instrument across subgroups. These findings are consistent with those reported in the original English version of FUNMOVES, suggesting that the psychometric properties of the tool are stable across linguistic and cultural contexts.

One of the objectives of this study was to assess the extent to which FUNMOVES and the Movement Assessment Battery for Children – Second Edition (MABC-2) identify the same individuals as having motor development difficulties. Given that the FUNMOVES assessment can be feasibly administered within a single Physical Education session by school-teachers, it is essential to evaluate its diagnostic accuracy.

When comparing FUNMOVES with the total score of the MABC-2, 11 false positives and three false negatives were identified. Nine of the 11 false positives scored below the 10th percentile in at least one MABC-2 subscale (Manual Dexterity, Aiming and Catching, or Balance). Regarding two of the false negatives, both scored at the 5th percentile in the Manual Dexterity subscale—a domain not specifically assessed by FUNMOVES—thereby influencing their overall MABC-2 scores. Notably, both individuals scored above the 15th percentile in the remaining two subscales.

A comparison of performance outcomes between FUNMOVES and each MABC-2 subscale revealed that FUNMOVES demonstrates a moderate ability to classify individuals according to their motor proficiency. Under the broad criteria, sensitivity was 0.67 (95% CI: 0.48–0.81) and specificity was 0.77 (95% CI: 0.57–0.90). This indicates balanced performance in identifying true positives and true negatives. The positive predictive value (PPV) was 0.78, and the negative predictive value (NPV) was 0.65, suggesting that FUNMOVES is more effective at confirming motor difficulties than ruling them out. The overall accuracy was 0.71 (95% CI: 0.58–0.82), supporting its potential as a practical school-based screening tool to aid in the early identification and referral of children who may require further motor assessment.

The ROC analysis showed that the FUNMOVES percentile provides moderate diagnostic accuracy (AUC ≈ 0.78) for identifying motor difficulties in primary-school children. This performance aligns with the upper band typically reported for school-based screening questionnaires pooled AUC ≈ 0.80 for the parent-completed Developmental Coordination Disorder Questionnaire (DCDQ) according to a recent meta-analysis [50] and is comparable to teacher tools such as the Motor

Observation Questionnaire for Teachers, which reaches an AUC of roughly 0.73 [51]. By contrast, very brief observational checklists like the original MABC teacher checklist offer high specificity but low sensitivity, yielding an effective AUC close to 0.55 [52]. These benchmarks suggest that FUNMOVES performs comparably to the best-validated school screeners. It clearly outperforms brief checklists, supporting its use as a practical first-line tool for early detection of motor impairment.

Using the strict criteria (focusing only on MABC-2 total score below the 10th percentile), FUNMOVES showed high sensitivity (0.80) but moderate specificity (0.68). This means the teacher-led screen identifies ~80% of children with motor difficulties, though it also falsely flags some children with normal motor skills (specificity 68%).

The broad criteria (defining MABC-2 positive as a total score below the 10th percentile or a score below the 10th percentile on either the Aiming & Catching or Balance subscale) shifted this balance: sensitivity dropped to 0.67, while specificity improved to 0.77. The broader definition captured approximately 67% of true cases but more accurately ruled out children without motor issues (77% specificity). Notably, both approaches yielded a similar overall accuracy (71%), yet their predictive values differed in line with the sensitivity-specificity trade-off. With the strict criterion, NPV was high (0.89), meaning children who pass the screen likely have age-appropriate motor skills. However, the positive predictive value was modest (PPV = 0.52), so nearly half of those flagged as "at-risk" by FUNMOVES did not meet the strict MABC motor difficulty threshold. The broad criteria yielded a higher PPV (0.78), meaning most flagged children had real motor issues. However, NPV dropped to 0.65, indicating a greater chance of missing children with actual difficulties.

These findings highlight the balance between sensitivity and specificity and its practical implications. A higher sensitivity (as seen with the strict criteria) is desirable in a school screening context to ensure that the majority of children with motor difficulties are identified for further assessment or support. The cost of this approach is a greater number of false positives, i.e., children who fail the screening but are not below criteria on the gold-standard MABC test. False positives can lead to unnecessary referrals or anxiety, which is a concern given limited healthcare and specialist resources (56). The optimal balance may depend on context: if the priority is early intervention for as many children as possible, a more sensitive (strict) threshold could be appropriate, whereas if resources for full assessments or interventions are very limited, a more specific (broad) threshold might be justified to focus on the most evident cases.

Importantly, broadening the reference definition to include MABC-2 subscales below the 10th percentile did not improve overall discrimination, suggesting that deficits captured at the subscale level are largely reflected in the total score. This simplifies clinical interpretation, as the strict criterion appears sufficient for concurrent validation of FUNMOVES. However, subtle shifts in classification were observed: under the broad criteria, 6 false positives became true positives, while 6 true negatives turned into false negatives. This suggests that although the total score captures most deficits, domain-specific difficulties may still influence diagnostic outcomes, consistent with literature describing heterogeneous DCD profiles [66].

Beyond its diagnostic accuracy, the implementation of FUNMOVES in schools offers significant systemic and social benefits. Schools are ideal for ecological assessment, where motor performance is observed in the child's natural, familiar environment. This approach not only enhances the validity of the assessment but also increases its accessibility and acceptability among educators and families. Moreover, PE lessons provide a particularly suitable context for this type of assessment, as FMS are already a core focus of PE curricula. Teachers are familiar with these skills and often seek better tools to assess them objectively, meaning that implementing FUNMOVES does not represent an additional burden but rather supports their existing educational goals.

In overburdened healthcare systems with long waiting lists, school-based tools like FUNMOVES can help reduce diagnostic delay. FUNMOVES enables early identification of children at risk for DCD in schools. It acts as a first filter, prioritizes and increases the likelihood of more children with motor difficulties getting 'fast-tracked' for more specialized evaluation. This triage approach would optimize healthcare resources and reduce the burden on diagnostic services. Additionally, compared to simpler screening tools such as parent- or teacher-completed questionnaires (e.g., the DCDQ), FUNMOVES provides a more formal and objective assessment. It is less prone to individual bias and offers greater equity, as it ensures that all children are assessed within the school setting, leading to higher response rates and more consistent data collection.

Furthermore, PE teachers participating in the study were asked to identify students they suspected might have motor difficulties. Interestingly, 6.7% of the children were flagged by teachers and included in the group with suspected difficulties. This proportion is not a random estimate; rather, it aligns closely with global prevalence estimates for DCD, reinforcing the relevance and potential accuracy of school-based identification methods like FUNMOVES. This finding also supports the ecological validity of the FUNMOVES tool, as it demonstrates its consistency with teachers' real-world observations in natural school settings, where early signs of motor difficulties are first detected.

Early detection is critical, as it allows for timely intervention during key developmental windows. Research consistently shows that early intervention is associated with better long-term outcomes in motor skills, academic performance, and psychosocial well-being [67–69]. Early support helps children develop adaptive strategies and improve participation in physical and social activities. It also reduces risks linked to undiagnosed DCD, such as low self-esteem, isolation, sedentary behavior, and health issues [68].

Moreover, the integration of FUNMOVES into the school system fosters a collaborative, multidisciplinary approach to child development. PE teachers, classroom tutors, subject-specific educators (particularly those focusing on fine motor skills), and school guidance professionals can collectively contribute to a more holistic understanding of each student's needs. This shared responsibility not only enhances the accuracy of referrals but also promotes a culture of inclusion and support within the educational community.

In summary, FUNMOVES is not only a valid and reliable screening tool for motor difficulties but also a strategic asset for early detection and intervention. Implementing FUNMOVES in schools may reduce underdiagnosis and shorten diagnostic timelines. It ensures early support, improving children's developmental outcomes and quality of life.

## 5 Conclusion

FUNMOVES has the potential to become a universal screening tool, freely accessible to schools, for detecting school-children with motor difficulties. This tool is robust, with moderate evidence of validity and reliability, enabling schools to support healthcare centers and families in identifying motor coordination disorders and difficulties. Additionally, this universal screening would allow for early intervention to reach children in collaboration with their families. Importantly, for FUNMOVES to be truly universal, it is essential to ensure its cross-cultural validity. While this is often taken for granted, explicitly validating the tool across different cultural contexts is crucial. Without such validation, there is a risk that, for example in Spain, a child might be flagged as having motor difficulties without sufficient evidence that the tool accurately reflects local developmental norms.

Moreover, FUNMOVES enables the rapid assessment of entire classes and large populations, offering a scalable approach that can deepen our understanding of the challenges children face in motor development across diverse educational settings. This broad application opens the door to exploring how motor difficulties intersect with other important factors such as academic performance, social participation, and lifestyle habits, ultimately contributing to more holistic and inclusive educational and health strategies.

## 6 Limitations and future proposals

Main limitations include a modest sample size and single cohort. Even with bootstrap correction, these factors may inflate performance estimates and limit generalizability to other age groups or settings. Future research should validate the ≤ 17th-percentile cut-off in independent samples and investigate whether combining FUNMOVES with contextual variables (e.g., perinatal history, physical-activity level) further enhances diagnostic precision.

Another limitation concerns the PSI values obtained in this study (0.53 initial, 0.52 rescored), which are below the conventional threshold of 0.70 typically required to distinguish multiple ability levels. This indicates that the scale may have limited precision for ranking individuals across a wide spectrum of motor competence. However, FUNMOVES was not designed for that purpose but rather as a screening tool to flag children who may present with motor difficulties. For

this binary distinction—identifying children at risk of pDCD versus those with typical motor development—the obtained PSI values are considered sufficient, particularly given the good item fit and the person–item distribution map supporting the instrument's capacity to detect lower motor proficiency. Future research should nevertheless aim to test the responsiveness of the tool in samples with a broader range of motor ability.

Future versions could integrate fine motor assessments, broadening the scope and improving the comprehensiveness of school-based screening. It is also important to consider the heterogeneity of DCD profiles, which may involve difficulties in specific motor domains not fully captured through physical education-based assessments. This highlights the value of interdisciplinary collaboration with other school professionals whose expertise may help identify challenges in areas that are less accessible through PE alone. Despite these caveats, the present findings support FUNMOVES as an effective and logistically feasible screening instrument that can facilitate early identification and timely referral of children with potential motor impairments.

## Supporting information

**S1 Fig. Category probability curves.** Fig2. a1) Running.
(TIF)

**S2 Fig. Category probability curves.** Fig2. a2) Jumping.
(TIF)

**S3 Fig. Category probability curves.** Fig2. a3) Hopping.
(TIF)

**S4 Fig. Category probability curves.** Fig2. a4) Throwing DOM.
(TIF)

**S5 Fig. Category probability curves.** Fig2. a5) Balance.
(TIF)

**S6 Fig. Category probability curves.** Fig2. b1) Running rescored.
(TIF)

**S7 Fig. Category probability curves.** Fig2. b2) Jumping rescored.
(TIF)

**S8 Fig. Category probability curves.** Fig2. b3) Hopping rescored.
(TIF)

**S9 Fig. Category probability curves.** Fig2. b4) Throwing rescored.
(TIF)

**S10 Fig. Category probability curves.** Fig2. b5) Balance rescored.
(TIF)

**S11 Fig. Fig 4. a) Broad criteria.**
(TIF)

**S12 Fig. Fig 4. b) Strict criteria.**
(TIF)

**S13 Fig. Fig 4. c) ROC curve.**
(TIF)

**S1 File. Teacher manual in Spanish.**
(PDF)

**S2 File. Teacher manual in English.**
(PDF)

**S3 File. Teacher response sheet in Spanish.**
(PDF)

**S4 File. Teacher response sheet in English.**
(PDF)

**S5 File. Database.**
(XLSX)

**S6 File. Code for concurrent and predictive validity analysis in R.**
(R)

## Author contributions

**Conceptualization:** Pablo Lizoain, Nick Preston, Lucy H. Eddy, Liam J. B. Hill, Sara Magallón.

**Data curation:** Pablo Lizoain, Martín Martínez.

**Formal analysis:** Pablo Lizoain, Martín Martínez.

**Funding acquisition:** Sara Magallón.

**Investigation:** Pablo Lizoain, Diana Rodriguez Romero, Celeste Reyes-Vivanco, Sara Magallón.

**Methodology:** Pablo Lizoain, Diana Rodriguez Romero, Celeste Reyes-Vivanco, Martín Martínez.

**Project administration:** Sara Magallón.

**Resources:** Pablo Lizoain, Nick Preston, Lucy H. Eddy, Liam J. B. Hill, Sara Magallón.

**Software:** Pablo Lizoain, Martín Martínez.

**Supervision:** Sara Magallón, Martín Martínez.

**Validation:** Pablo Lizoain, Sara Magallón, Martín Martínez.

**Visualization:** Pablo Lizoain, Sara Magallón, Martín Martínez.

**Writing – original draft:** Pablo Lizoain, Sara Magallón, Martín Martínez.

**Writing – review & editing:** Pablo Lizoain, Diana Rodriguez Romero, Celeste Reyes-Vivanco, Nick Preston, Lucy H. Eddy, Liam J. B. Hill, Sara Magallón, Martín Martínez.

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
