## [Decision Letter · Decision Letter 0]

20 Oct 2025

Dear Dr. Lizoain,

Thank you for submitting your manuscript to PLOS ONE. After careful consideration, we feel that it has merit but does not fully meet PLOS ONE’s publication criteria as it currently stands. Therefore, we invite you to submit a revised version of the manuscript that addresses the points raised during the review process.

We look forward to receiving your revised manuscript.

Kind regards,

Jindong Chang, Ph.D.

Academic Editor

PLOS ONE

Journal Requirements:

“P. Lizoain is supported by the grant PRE2021-097858, funded by MCIN/AEI/ 10.13039/501100011033 and by ESF+ U. L.H. Eddy is supported by a grant from the Waterloo Foundation (ref: 27665413). S. Magallón is supported by the Ramón y Cajal grant RYC-2017-22060 funded by MCIN/AEI/ and by PID2020–119328GA-I00 AEI Proyectos I + D + i funded by MCIN/AEI.” 

3. Please expand the acronym “MCIN” (as indicated in your financial disclosure) so that it states the name of your funders in full.

4. Please note that funding information should not appear in any section or other areas of your manuscript. We will only publish funding information present in the Funding Statement section of the online submission form. Please remove any funding-related text from the manuscript.

5. We notice that your supplementary figures are uploaded with the file type 'Figure'. Please amend the file type to 'Supporting Information'. Please ensure that each Supporting Information file has a legend listed in the manuscript after the references list.

6. We notice that your supplementary figures 1-3 and tables 1-6 are included in the manuscript file. Please remove them and upload them with the file type 'Supporting Information'. Please ensure that each Supporting Information file has a legend listed in the manuscript after the references list.

7. We note that there is identifying data in the Supporting Information file Database - PAPER FUNMOVES.xlsx. Due to the inclusion of these potentially identifying data, we have removed this file from your file inventory. Prior to sharing human research participant data, authors should consult with an ethics committee to ensure data are shared in accordance with participant consent and all applicable local laws.

-Location data

Data that are not directly identifying may also be inappropriate to share, as in combination they can become identifying. For example, data collected from a small group of participants, vulnerable

populations, or private groups should not be shared if they involve indirect identifiers (such as sex, ethnicity, location, etc.) that may risk the identification of study participants.

Please remove or anonymize all personal information ID and age, ensure that the data shared are in accordance with participant consent, and re-upload a fully anonymized data set. Please note that spreadsheet columns with personal information must be removed and not hidden as all hidden columns will appear in the published file.

Reviewers' comments:

Reviewer's Responses to Questions

**Comments to the Author**

1. Is the manuscript technically sound, and do the data support the conclusions?

Reviewer #1: Yes

2. Has the statistical analysis been performed appropriately and rigorously?

Reviewer #1: Yes

3. Have the authors made all data underlying the findings in their manuscript fully available?

Reviewer #1: Yes

4. Is the manuscript presented in an intelligible fashion and written in standard English?

Reviewer #1: Yes

Reviewer #1: Title:

Validation of FUNMOVES: A reliable tool for assessing motor skills in Spanish schoolchildren

Abstract:

Aim: "evaluate its psychometric properties and effectiveness" – "Effectiveness" is vague. Consider using "diagnostic accuracy" or "criterion validity" which is more precise for this context.

Results: "Using the MABC-2 10th percentile as the criterion" – This contradicts the main text where the "strict" criterion for the main analysis is the 15th percentile. The abstract must accurately reflect the primary outcome reported in the paper. Verify which criterion was primary.

Page 11, Line 113-115: The justification for using MABC-2 over MABC-3 is sound but should include a citation for the claim that MABC-3 is not yet validated in Spain.

Page 15, Table 2: The column "Suspected difficulties" has values for "Yes" but no corresponding percentage values in the total row, unlike the other columns. This is a minor formatting inconsistency.

Page 15, Participant Flow It would be helpful to include a participant flow diagram (e.g., CONSORT-style) in the main text or supplementary materials to clearly show the recruitment, exclusion, and group allocation process.

Page 17, Table 3: The modification for "Balance" says "(considered dropping category 1)." Was category 1 dropped or not? The text says categories 3 and 4 were combined. This should be clarified.

Page 20, Table 5: The p-value for "Suspected difficulties" is 0.011, indicating a significant difference between TD and pDCD groups in teacher suspicion. This is an interesting and expected finding that could be briefly mentioned in the text.

Page 20, Fig 3 Caption: Ensure the final figures have clear, legible axes and labels.

Page 21: The discussion of false positives/negatives is excellent and provides crucial context for interpreting the tool's performance.

Page 22: The paragraph on teacher suspicion (6.7% prevalence) is a strong point that reinforces the tool's ecological validity.

Page 24, Limitations: The limitation regarding the lack of fine motor assessment is well-noted. The suggestion for future interdisciplinary collaboration is a valuable addition.

The final sample for the main analysis (n=243) is adequate for Rasch analysis. However, the subsample for validation against MABC-2 is small (n=50, with n=24 in the pDCD group). This limits the precision of the diagnostic accuracy metrics (wide confidence intervals) and the generalizability of the proposed cut-off points.

The PSI values of 0.53 (initial) and 0.52 (rescored) are reported as "acceptable." However, a PSI < 0.7 is typically considered to have low reliability for separating individuals into more than two groups (e.g., [57,58]). The authors correctly note it is suitable for identifying low proficiency (a binary distinction), but this requires clearer justification.

The terminology "Strict" and "Broad" is potentially misleading. Strict is defined as MABC-2 Total < 15th percentile (not the standard 5th or 10th for diagnosis), which is actually a *lenient* clinical criterion. Broad is defined as Total < 10th *or* subscale < 10th, which is a different, more inclusive approach. Clarify the rationale for choosing the 15th percentile as the "strict" criterion. Consider using more descriptive labels like "Total Score Criterion (15th %ile)" and "Total or Subscale Criterion (10th %ile)" to avoid value-laden terms like "strict" and "broad."

The main manuscript text references tables and figures (e.g., Table 1, Fig 1, Fig 2, Fig 3) that are described in the text but are not included in the provided PDF pages. The PDF contains placeholder text and formatting instructions for them (e.g., "Fig 1:", "Table 4 – Descriptive statistics...").

The Data Availability Statement on page 9 states data will be available "upon acceptance." However, the response on page 7 states "All relevant data are within the manuscript and its Supporting Information files." These statements conflict.

**Do you want your identity to be public for this peer review?** For information about this choice, including consent withdrawal, please see our Privacy Policy

Reviewer #1: **Yes: ** Amir Shams, Sport Sciences Research Institute of Iran

---

## [Author Response · Author response to Decision Letter 1]

31 Oct 2025

Academic Editor (E)

E.C0. After careful consideration, we feel that it has merit but does not fully meet PLOS ONE’s publication criteria as it currently stands. Therefore, we invite you to submit a revised version of the manuscript that addresses the points raised during the review process.

E.R0. We sincerely thank the Academic Editor for recognizing the merit of our study and for providing us with the opportunity to revise the manuscript.

We have carefully addressed all the editorial and reviewer comments, and we believe that the revised version now fully meets PLOS ONE’s publication standards.

We hope that the changes implemented improve the clarity and completeness of the manuscript.

E.C1. Please ensure that your manuscript meets PLOS ONE's style requirements, including those for file naming.

E.R1. We thank the Academic Editor for this reminder. We have carefully reviewed the manuscript and all uploaded files to ensure full compliance with PLOS ONE’s style and file-naming requirements. The filenames have been standardized according to the journal’s guidelines.

Revised filenames:

- Figures: Fig1 to Fig4 are uploaded as figures and the caption is mentioned in the manuscript.

- Supplementary material: S1_Fig.tif to S13_Fig.tif and S1_File to S6_File

We confirm that all files now follow the prescribed format.

E.C2. Please state what role the funders took in the study. If the funders had no role, please state: "The funders had no role in study design, data collection and analysis, decision to publish, or preparation of the manuscript." If this statement is not correct you must amend it as needed.

E.R2. We have now included the statement recommended by PLOS ONE to clarify the role of the funders. The Funding Statement in the manuscript and the cover letter have been updated as follows:

We confirm that this statement accurately reflects the role of the funding bodies in our study.

E.C3. Please expand the acronym “MCIN” (as indicated in your financial disclosure) so that it states the name of your funders in full. This information should be included in your cover letter; we will change the online submission form on your behalf.

E.R3. We have now expanded the acronym “MCIN” to indicate the full name of the funding bodies in both Spanish and English. The revised Funding Statement specifies the individual grants received by the authors and now reads as follows:

“Several authors received external funding supporting their research contracts:

P. Lizoain is supported by the grant PRE2021-097858, funded by the Ministerio de Ciencia e Innovación (MCIN) / Ministry of Science and Innovation (Spain) and the Agencia Estatal de Investigación (AEI) / State Research Agency (Spain), and co-funded by the European Social Fund Plus (ESF+ U): MCIN/AEI/ 10.13039/501100011033.

L.H. Eddy is supported by a grant from the Waterloo Foundation (ref: 27665413).

S. Magallón is supported by the Ramón y Cajal grant RYC-2017-22060, funded by the Ministerio de Ciencia e Innovación (MCIN) / Ministry of Science and Innovation (Spain) and the Agencia Estatal de Investigación (AEI) / State Research Agency (Spain), and by the project PID2020–119328GA-I00 (AEI Proyectos I+D+i, funded by MCIN/AEI).

The funding sources had no role in study design, data collection and analysis, decision to publish, or preparation of the manuscript.”

The same information has been included in the cover letter so that the submission form can be updated accordingly.

E.C4. Please note that funding information should not appear in any section or other areas of your manuscript. We will only publish funding information present in the Funding Statement section of the online submission form. Please remove any funding-related text from the manuscript.

E.R4. We have carefully reviewed the entire manuscript, including all sections and supplementary materials, to ensure that no funding information appears outside the designated Funding Statement.

We confirm that all mentions of financial support are now limited to the Funding Statement section, in full accordance with PLOS ONE’s editorial policy.

E.C5. We notice that your supplementary figures are uploaded with the file type 'Figure'. Please amend the file type to 'Supporting Information'. Please ensure that each Supporting Information file has a legend listed in the manuscript after the references list.

E.R5. We have reviewed all uploaded figures and amended the file types to “Supporting Information” as requested. Legends for each supplementary figure have now been included in the manuscript, listed after the References section.

List of Supporting Information legends (as included in the revised manuscript):

SUPPORTING INFORMATION

S1 Fig. Category Probability Curves. Fig1. a1) Running

S2 Fig. Category Probability Curves. Fig1. a2) Jumping

S3 Fig. Category Probability Curves. Fig1. a3) Hopping

S4 Fig. Category Probability Curves. Fig1. a4) Throwing DOM

S5 Fig. Category Probability Curves. Fig1. a5) Balance

S6 Fig. Category Probability Curves. Fig1. b1) Running rescored

S7 Fig. Category Probability Curves. Fig1. b2) Jumping rescored

S8 Fig. Category Probability Curves. Fig1. b3) Hopping rescored

S9 Fig. Category Probability Curves. Fig1. b4) Throwing rescored

S10 Fig. Category Probability Curves. Fig1. b5) Balance rescored

S11 Fig. Fig 3. a) Broad criteria

S12 Fig. Fig 3. b) Strict criteria

S13 Fig. Fig 3. c) ROC curve

S1 File. Teacher manual in Spanish.

S2 File. Teacher manual in English.

S3 File. Teacher response sheet in Spanish.

S4 File. Teacher response sheet in English.

S5 File. Database.

S6 File. Code for concurrent and predictive validity analysis in R.

We confirm that all supplementary materials now conform to PLOS ONE’s formatting and labeling requirements.

E.C6. We notice that your supplementary figures 1-3 and tables 1-6 are included in the manuscript file. Please remove them and upload them with the file type 'Supporting Information'. Please ensure that each Supporting Information file has a legend listed in the manuscript after the references list.

E.R6. We thank the Academic Editor for this clarification and for pointing out the correct procedure regarding supplementary materials. We realized that, in the initial submission, we had misunderstood the required format for including figures and tables. In the revised version, we have now ensured that:

● Only the main figures and tables remain at the manuscript file as required.

● Supplementary figures (S1–S13) and files (S1–S6) have been uploaded separately as Supporting Information.

● Legends for all supplementary materials are now listed after the References section in the manuscript.

We believe that the presentation of figures and tables now fully complies with PLOS ONE’s formatting and file submission requirements. This comment is related to R1.C12.

E.C7. Please remove or anonymize all personal information ID and age, ensure that the data shared are in accordance with participant consent, and re-upload a fully anonymized data set. Please note that spreadsheet columns with personal information must be removed and not hidden as all hidden columns will appear in the published file.

E.R7. We have reviewed the shared dataset to ensure full compliance with PLOS ONE’s anonymization and data protection requirements.

Participant IDs were already fully anonymized (B001–B243) and contain no personal information. The dataset does not include any columns with potentially identifying details, and all non-essential variables have been removed rather than hidden.

The variable “age” has been retained because it is essential for several analyses, but it cannot lead to participant identification as no dates of birth or assessment are included.

The anonymized dataset has been confirmed and uploaded as S5_File.xlsx

E.C8. If the reviewer comments include a recommendation to cite specific previously published works, please review and evaluate these publications to determine whether they are relevant and should be cited. There is no requirement to cite these works unless the editor has indicated otherwise.

E.R8. We thank the Academic Editor for this clarification. We will carefully evaluate the publications suggested by the reviewer to determine their relevance to our study.Where appropriate, we will incorporate the recommended citations to strengthen the manuscript and ensure full contextual accuracy.

Specifically, we plan to review the reference suggested regarding the validation status of the MABC-3 in Spain and include it if deemed pertinent.

Reviewer 1 (R1)

1. Is the manuscript technically sound, and do the data support the conclusions?

Reviewer #1: Yes

2. Has the statistical analysis been performed appropriately and rigorously?

Reviewer #1: Yes

3. Have the authors made all data underlying the findings in their manuscript fully available?

Reviewer #1: Yes

4. Is the manuscript presented in an intelligible fashion and written in standard English?

Reviewer #1: Yes

5. Review Comments to the Author

Reviewer #1: Title:

Validation of FUNMOVES: A reliable tool for assessing motor skills in Spanish schoolchildren

R1.C1 Abstract:

Aim: "evaluate its psychometric properties and effectiveness" – "Effectiveness" is vague. Consider using "diagnostic accuracy" or "criterion validity" which is more precise for this context.

Results: "Using the MABC-2 10th percentile as the criterion" – This contradicts the main text where the "strict" criterion for the main analysis is the 15th percentile. The abstract must accurately reflect the primary outcome reported in the paper. Verify which criterion was primary.

R1.R1. We thank the reviewer for this valuable suggestion.

In the Abstract, we have replaced the term “effectiveness” with “diagnostic accuracy” to better capture the aim and scope of the study.

We have also revised the description of the results to clarify the reference criterion used, which now reads: “Using the MABC-2 10ᵗʰ percentile in total score as the criterion for motor impairment.”

This adjustment maintains consistency with the strict criterion definition presented in the main text, where the distinction between Strict (only Total Score) and Broad (Total Score and subscales) criteria is fully explained (see response to Comment R1.C11).

R1.C2. Page 11, Line 113-115: The justification for using MABC-2 over MABC-3 is sound but should include a citation for the claim that MABC-3 is not yet validated in Spain.

R1.R2. We thank the reviewer for this helpful comment.

We have now added a reference [45] to support our statement regarding the use of the MABC-2 instead of the MABC-3. The justification has been updated to indicate that the MABC-3 has not yet been validated or adapted for use in Spain.

R1.C3. Page 15, Table 2: The column "Suspected difficulties" has values for "Yes" but no corresponding percentage values in the total row, unlike the other columns. This is a minor formatting inconsistency.

R1.R3. We thank the reviewer for noticing this formatting inconsistency. The percentage values in the “Suspected difficulties” column were revised to ensure consistency across all variables in Table 2. The corrected percentages now explicitly reflect the proportion of participants identified as “Yes” and “No,” excluding those not grouped under either category.

The total percentage for the “Yes” category has been added and now reads 6.59%. The percentage for the “No” category is 91.77.

This change affects only the presentation of results and does not modify any of the analyses or interpretations reported in the manuscript.

R1.C4. Page 15, Participant Flow It would be helpful to include a participant flow diagram (e.g., CONSORT-style) in the main text or supplementary materials to clearly show the recruitment, exclusion, and group allocation process.

R1.R4. We thank the reviewer for this excellent suggestion.

We have now added a participant flow diagram (Figure 1) to clearly illustrate the recruitment, exclusion, and selection processes for both the FUNMOVES and MABC-2 subsamples. This figure follows a CONSORT-style format and summarizes the key stages, including the number of participants invited, those who provided informed consent, exclusions (e.g., absence, DSM-5 criterion D, administration issues), and the final MABC-2 subsample.

Figure 1. Participant flow diagram. Flow diagram showing the recruitment process, exclusions, and formation of the FUNMOVES and MABC-2 subsamples.

The addition of this figure required the renumbering of subsequent figures throughout the manuscript.

A reference to this figure has also been included in the Methods section for clarity.

We believe that the inclusion of this diagram enhances the transparency and readability of the study design.

R1.C5. Page 17, Table 3: The modification for "Balance" says "(considered dropping category 1)." Was category 1 dropped or not? The text says categories 3 and 4 were combined. This should be clarified.

R1.R5. We thank the reviewer for pointing out this ambiguity. The phrase “considered dropping category 1” was removed to avoid confusion. After reviewing the scoring process, we decided to retain all categories for the Balance subscale. Only categories 3 and 4 were combined to optimize model fit, while maintaining the qualitative differentiation among performance levels, particularly in the lower ability range.

This revision clarifies that no category was dropped, ensuring both conceptual consistency and data integrity in Table 3 and the related text.

R1.C6. Page 20, Table 5: The p-value for "Suspected difficulties" is 0.011, indicating a significant difference between TD and pDCD groups in teacher suspicion. This is an interesting and expected finding that could be briefly mentioned in the text.

R1.R6. We thank the reviewer for this insightful observation. We have now incorporated a sentence in the Results section (page 20, immediately after Table 5) to highlight the significant difference between groups regarding teacher suspicion (p = 0.011). The added text reads:

“In addition, a significant difference was observed between groups in the variable teacher suspicion (p = 0.011), with a higher proportion of suspected cases in the pDCD group. This result was expected, as children classified as pDCD based on FUNMOVES (scores below the 10ᵗʰ percentile or FUNMOVES scores below the 15ᵗʰ percentile plus teacher suspicion) are more likely to have been previously identified because of the definition of the group pDCD.”

This addition provides context for the finding and connects it to the classification logic underlying both the FUNMOVES and MABC-2 assessments.

R1.C7. Page 20, Fig 3 C

---

## [Editor Report · Decision Letter 1]

11 Nov 2025

Validation of FUNMOVES: A reliable tool for assessing motor skills in Spanish schoolchildren

PONE-D-25-35822R1

Dear Dr. Lizoain,

We’re pleased to inform you that your manuscript has been judged scientifically suitable for publication and will be formally accepted for publication once it meets all outstanding technical requirements.

Kind regards,

Jindong Chang, Ph.D.

Academic Editor

PLOS ONE
---

## [Editor Report · Acceptance letter]

PONE-D-25-35822R1

PLOS ONE

Dear Dr. Lizoain,

I'm pleased to inform you that your manuscript has been deemed suitable for publication in PLOS ONE. Congratulations! Your manuscript is now being handed over to our production team.

Kind regards,

on behalf of

Dr. Jindong Chang

Academic Editor

PLOS ONE